# Factors Affecting Internal Audit Effectiveness: Empirical Evidence from Vietnam

**Thu Trang Ta and Thanh Nga Doan** *

School of Accounting and Auditing, National Economics University, Hanoi 10000, Vietnam; trangtt@neu.edu.vn
* Correspondence: doanthanhnga@neu.edu.vn; Tel.: +84-906290208

**Abstract:** This study investigated four factors affecting internal audit effectiveness in Vietnam, namely, independence of internal audit, the competence of internal auditors, management support for internal audit, and quality of internal audit work. Quantitative and qualitative evaluations were conducted, including a logistics regression model and other analyses, using SPSS software. Through semi-structured in-depth interviews and an online survey, 144 responses were obtained from internal Vietnamese auditors of nonfinancial companies listed on the Vietnamese stock market in 2021. After processing the data, the results revealed two factors (independence of internal auditor and management support for internal audit) with a positive influence on internal audit effectiveness, whereas the competence of internal auditors and quality of internal audit work did not affect internal audit effectiveness.

**Keywords:** effectiveness; internal audit; Vietnam

## 1. Introduction

Changes in the economy and political environment, along with advances in information technology, have significantly affected corporate governance. Enterprises are facing many business risks, leading to many different types of financial fraud in recent years. The detection and prevention of business risks and large-scale financial fraud pose several challenges for boards of directors and senior management, with respect to internal audit activities. One of the risks faced by organizations is data protection (Pavelek and Zajíčková 2021). To eliminate possible risks and improve the performance of activities, organizations should perform organizational audits (Mucha 2021), particularly internal audits. In order to strengthen the effectiveness of internal audits, their characteristics should be investigated in the framework of the legal system and professional standards. In particular, following the economic collapse in the face of fraudulent financial reporting in the 20th century, government agencies and professional associations reformed the legal system to enhance internal audit effectiveness in corporate governance, encompassing risk management and control processes. According to the IPPF (2017), internal auditing is an independent, objective assurance and consulting activity designed to add value to and improve an organization's operations. It supports an organization in accomplishing its objectives through a systematic approach and follows an appropriate implementation process to assess and enhance the effectiveness of internal control, risk management, and governance processes. Internal audit effectiveness is associated with the way in which an internal audit helps an organization obtain its goals and creating added value for activities in various aspects. Due to the important role of internal audits in an organization, their effectiveness has been the subject of research around the world (Mihret and Yismaw 2007; Ahmad et al. 2009; Arena and Azzone 2009; Baharud-din et al. 2014; Alzeban and Gwilliam 2014; Sakour and Laila 2015; Dellai and Omri 2016). However, studies have typically focused on factors affecting internal audit effectiveness in the public and banking sectors, whereas the research on internal audit effectiveness in listed companies is limited.

The capital market includes financial institutions to mobilize medium- and long-term capital for economic entities through investing capital on the stock market (Peracek 2020). The stock market is the most important part of the capital market for economic entities and an important capital mobilization channel for the economy. With 25 years of operation so far, Vietnam's stock market has about 747 listed companies, among which the Ho Chi Minh City Stock Exchange (HOSE) has 404 listed companies and the Hanoi Stock Exchange (HNX) has 343 listed companies. As of 15 October 2021, the VN Index reached 1392.7 points, an increase of 573.4% compared with the end of 2000. The stock market capitalization reached VND 7,166,000 billion, an increase of 626.8% compared with the end of 2000, equivalent to 113.9% of the GDP (Nguyen 2021).

However, the study of internal audit effectiveness in listed companies has several limitations because many listed companies lack an understanding of the function and responsibilities of an internal auditor, as well as of how to support their operations, in compliance with Decree 05/2019/ND-CP of Vietnamese Governance before 1 April 2021.

In the literature, there has been little research regarding internal audits of nonfinancial listed companies. Therefore, the overall objective of this paper was to research the factors affecting internal audit effectiveness in nonfinancial companies listed on Vietnam's stock market.

## 2. Literature Review and Hypotheses

Previous studies have investigated the factors with an influence on internal audit effectiveness. Firstly, internal audit effectiveness is based on the internal audit concept of IIA (Mihret et al. 2010; Gros et al. 2017). The important objective of an internal audit is to provide an independent and objective auditing and advisory function, designed to create added value and complete organizational activities. An internal audit helps an organization to achieve its goals by adopting a systematic approach to assess and improve the effectiveness of risk management processes, control processes, and corporate governance (Hass et al. 2006; Yee et al. 2008; Walter and Guandaru 2012; Dellai and Omri 2016; IPPF 2017). In particular, Yee et al. (2008) conducted a study interviewing 83 Singaporean managers, including junior, middle, and senior managers, to investigate their viewpoints regarding the role of an internal audit and its effectiveness. The results indicated that both senior and junior managers in Singapore considered the role of internal auditors in the business to be important. In addition, the perspective of internal audit clients has been investigated in terms of the role and effectiveness of an internal audit. Dellai and Omri (2016) tested the factors affecting internal audit effectiveness by collecting 148 responses from chief audit executives working for organizations in Tunisia. This study revealed several measures of internal audit effectiveness, including the establishment of an audit plan, issue of internal audit reports, performance of audit recommendations, and assessment and improvement of activities in compliance with IIA standards.

Secondly, factors which affect internal audit effectiveness include the independence of the auditor and the department of the internal audit, the competence of internal auditors, support for the internal audit department from managers, and the quality of audit work. Mutchler (2003) and Al-Akra et al. (2016) have pointed out that an important factor affecting the effectiveness of an internal audit is the independence of the internal audit function and auditors. Dellai and Omri (2016) indicated that the independence of the internal auditor has a positive influence on the effectiveness of the internal audit. The competence of auditors in the audit department is the most important factor affecting the effectiveness of all internal audit activities (Al-Twaijry et al. 2003; Alzeban and Gwilliam 2014). The support for the internal audit department from the manager is a key factor in accepting and improving the role of the internal audit department in the business (Ahmad et al. 2009; Alzeban and Gwilliam 2014; Dellai and Omri 2016). In particular, Alzeban and Gwilliam (2014) interviewed 203 managers and 239 internal auditors from 79 public sector organizations to evaluate factors affecting internal audit effectiveness in Saudi Arabia. These results showed the relationship between support from management and effectiveness of internal

audit from public organizations. In addition, Dellai and Omri (2016) have demonstrated a positive relationship between management support with internal audit effectiveness. Mihret and Yismaw (2007); Alzeban and Gwilliam (2014) have shown the impact of the quality of audit work on the effectiveness of internal audits.

From an overview of previous studies, it can be seen that there are four main factors affecting the effectiveness of an internal audit, shown subsequently.

### 2.1. Independence of Internal Audit and Effectiveness of Internal Audit Function

Independence is the freedom from situations and relationships that threaten internal auditor's ability to objectively perform responsibilities of internal audits in an unbiased manner.

In order to achieve a degree of independence for the internal audit function, the chief audit executive has the right to direct and unrestricted access to senior management and those charged with governance to communicate and report. Especially, reporting to a level within the organization must be performed by the chief audit executive to fulfill internal audit function and responsibilities. The chief audit executive must report the independence of the internal audit to the board, at least annually. Independence is a factor that performs its functions without facing conditions that can lead to an impartiality of internal auditors exercising their responsibilities (IPPF 2017). Independence is one of the mandatory requirements in the system of professional ethical standards for auditors. However, due to the characteristic of the organizational structure of internal auditors in enterprises, the internal audit department and internal auditors are an important part of the organizational structure. Internal audit department and the staff of the internal audit department need to be independent from other departments within the entity such as senior management, head of other departments to ensure implementing and giving opinions effectively. These relationships need to be studied to evaluate their influence on the effectiveness of internal audits. Mutchler (2003); Baharud-din et al. (2014); and Al-Akra et al. (2016) have pointed out that the independence of internal audit departments and internal auditors is an important factor affecting the effectiveness of internal audits. In order to measure the independence of internal audits, the IPPF (2017) has introduced requirements for the reporting function of internal audits, including administrative reports to senior management and the functional reporting of professional responsibilities to the Board of Director/Audit Committee or the Board Controller. Internal auditors have the right to unrestricted access to documents, personnel, and other departments in collecting audit evidence, avoiding conflicts of interest between internal auditors with other departments and senior management (Goodwin and Yeo 2001; Christopher et al. 2009). In addition, Alizadeh (2011) pointed out that the independence of internal audits is one of five factors that plays an important role in affecting the effectiveness of internal audits. Soh and Martinov-Bennie (2011) and Alzeban and Gwilliam (2014) think that the position of the internal audit department in the organization and relationship with the Board or Director (Audit Committee) or the senior management has a positive relationship with the effectiveness of internal audits. Abdolmohammadi (2009) and Arena and Azzone (2009) have presented that in order to increase the effectiveness of internal auditing, the Chief Audit Executives must be members of the Internal Audit Institute of the United States. Becoming members of the Internal Audit Institute ensures that internal auditors strictly comply with the requirements of professional ethical principles, especially the independence of internal auditors. According to previous studies and research, the authors established the first research hypothesis for this study:

**Hypothesis 1 (H1).** *Independence of the internal audit has a positive influence on the effectiveness of the internal audit.*

### 2.2. Competence of Internal Auditors and the Effectiveness of Internal Audit Functions

The IPPF (2017) also emphasizes the importance of the professional competence of internal auditors. The knowledge, skills and other competencies are important and necessary factors affecting the professional competence of internal auditors to perform the functions of internal audits. It assists auditors in assessing the client's current business conditions and emerging issues, thereby providing advice and recommendations to improve the organizational activities. Internal auditor competence is demonstrated by obtaining appropriate professional certifications and qualifications.

The competence of internal auditors is the most important factor affecting the effectiveness of all internal audit activities (Al-Twaijry et al. 2003; Alzeban and Gwilliam 2014). Previous studies pointed out that competence of internal auditors is an important variable affecting internal auditor effectiveness. Al-Matarneh (2011); Mihret and Woldeyohannis (2008); and Ali and Owais (2013) suggested that measurements of the competence of internal audits are based on necessary education, professional qualifications, experience and the number of training hours of an internal auditor, as well as the continued updating of professional training. Moreover, Fanning and Piercey (2014) pointed out that soft skills such as communication, persuasion and interaction skills with organizational divisions are important measures affecting the effectiveness of internal audits. Al-Twaijry et al. (2003) have confirmed that the effectiveness of internal auditors has been positively influenced by adequate levels of internal auditors' competence in training, experience, knowledge and professional qualifications. Other researchers found that the lack of competence of internal auditor is a problem that negatively affects the effectiveness of internal audits in some African countries (Mihret and Yismaw 2007; Onumah and Krah 2012; Walter and Guandaru 2012). Based on previous studies, the following hypothesis is designed:

**Hypothesis 2 (H2).** *The competence of internal auditors has a positive influence on the effectiveness of the internal audit.*

### 2.3. Management Support for Internal Audits and the Effectiveness of Internal Audit Functions

Strakova et al. (2021) demonstrated that managers of enterprises increasingly look for new methods to manage and run their business. The support and commitment of management also are important factors to improve the effectiveness of internal audits. Strong support of management for the work of internal auditors and the audit process makes for the success of the internal audit function. It is accepted that the internal audit process is an important activity in comparison with any other process within the organization. Some researchers indicated that with a lack of management approval, support and encouragement, the internal audit process is likely to be a time-consuming failure and a waste of money. The support of the internal audit department from the manager is a key element in accepting and appreciating the role of the internal audit department in business. In the IPPF (2017), the internal audit department needs to have the support of senior management and the Board of Directors to perform the duties and responsibilities in overseeing the control and risk management, and corporate governance in enterprise. Mihret et al. (2010) conducted research on public organizations in the United States. The results of the research showed that management support is the most important impact influencing the effectiveness of internal audits. Gramling et al. (2004); Salameh et al. (2011); Ahlawat and Lowe (2004); Aikins (2011); and Octavia (2013) showed that a positive attitude of managers towards internal audits plays an important role in the audit planning process and the effectiveness of internal audits. Ahmad et al. (2009), in their research on internal audits in the public sector and local government of Malaysia, pointed out that support from management is a factor that influences the implementation of audit recommendations, the number of auditors and the appropriate budget for the implementation of audit processes and the development of internal audit functions. Dellai and Omri (2016) demonstrated the positive influence of management support for internal audits. The Board of Directors will support the approval of internal audit regulations and plans, appointments, resigning

and salary decisions for internal auditors. For managers, support for internal audits in the administrative approval of personnel and allocated budgets for internal auditors to implement the audit plan, actively supports the audit department internally in training and professional development. In addition, senior management should be aware of the importance of the internal audit's role in the organization and support the internal audit's compliance with the laws, responsibilities, and authority of internal auditors.

Internal audit departments must ensure that appropriate, sufficient internal audit resources and the effective operation of internal audits can achieve the approved plan. As such, Onumah and Krah (2012) found that the lack of management support and insufficient human resources in the internal audit function mainly led to the ineffectiveness of the internal audit. In addition, Alzeban and Gwilliam (2014) found that the support of management has a strong relationship with the three auditing effectiveness dimensions. Therefore, the third hypothesis is established as follows:

**Hypothesis 3 (H3).** *Management support for internal audits has a positive influence on the effectiveness of the internal audit.*

*2.4. Quality of Internal Audit Work and Effectiveness of Internal Audit Function*

Cohen and Sayag (2010) and Dejnaronk et al. (2016) showed that performing an audit in accordance with internal audit standards affected the effectiveness of the internal audit. Mihret and Yismaw (2007) performed research which examined the impact of internal audit quality on the effectiveness of internal audits. These studies showed that the quality of an internal audit influences the effectiveness of the internal audit. Cohen and Sayag (2010) pointed out that the quality of the audit was evaluated by the annual audit plan, the areas audited and regular follow-up with an internal audit; thus, the following hypothesis was developed:

**Hypothesis 4 (H4).** *The quality of an internal audit positively influences the internal audit effectiveness.*

**3. Research Methodology**

After reviewing previous studies related to the factors affecting the effectiveness of internal audits, the authors pointed out four factors that significantly impact the effectiveness of internal audits by quantitative and qualitative research methods and measured these factors through appropriate observed variables and established an appropriate research model.

Based on previous studies by Goodwin and Yeo (2001); Mutchler (2003); Christopher et al. (2009); Cohen and Sayag (2010); Walter and Guandaru (2012); Baharud-din et al. (2014); Alzeban and Gwilliam (2014); Dellai and Omri (2016) and Al-Akra et al. (2016), the author used regression to assess the effect of independent variables on internal auditors' effectiveness. An illustration of a regression model is shown in Equation (1).

$$HL = \beta_0 + \beta_1 DL + \beta_2 NL + \beta_3 HT + \beta_4 CL + \varepsilon_i \tag{1}$$

The authors measured dependent variables and independent variables as follows:

Dependent variable: HL—internal audit effectiveness is measured by twelve observed variables: HL1 (internal audit is conducted in compliance with the audit chapter and audit manual, legal requirement of company); HL2 (the engagement results are communicated in a timely manner by the internal audit); HL3 (appropriate recommendations of an internal audit improve the process of organization); HL4 (a follow-up process are conducted by internal audit to confirm that corrective actions have been effectively conducted); HL5 (the effectiveness and efficiency of business operations and programs are reviewed by an internal audit); HL6 (reliable financial information and the integrity of financial reporting are reviewed by internal audit); HL7 (conformation with procedures, policies, plans and regulations are reviewed by internal audit); HL8 (internal audit evaluates that missions of organizational are agreed with their objectives); HL9 (internal audit evaluates the fraud

risks and how the organization responds and manages to fraud risk); HL10 (internal audit evaluates and improves e-governance process effectiveness); HL11 (internal audit evaluates and improves risk management effectiveness; HL12 (internal audit evaluates and improves internal control process effectiveness).

Independent variables include:

- DL (independence of internal audit) is measured by nine observed variables: DL1 (internal auditor has a lack of independence to conduct their professional obligations and duties); DL2 (the chief audit executive reports to senior management and the board that the internal audit has the right to fulfill its responsibilities); DL3 (the chief audit executive has no right to communicate directly with the board); DL4 (the internal audit department has no right directly to communicate directly with senior management other than the finance director); DL5 (conflicts of interest are rarely present in the internal auditors' work); DL6 (management has little interference in internal auditors' work); DL7 (internal auditors have no right to freely access all human resources and organization's departments); DL8 (the appointment and replacement of the chief audit executive are approved by those charged with governance); DL9 (internal auditors are not required to perform non-assurance activities).

- NL (the competence of internal auditor) is measured by nine observed variables: NL1 (internal auditors have enough appropriate experience to understand and evaluate the systems of the organization); NL2 (the internal auditors have appropriate knowledge and skills in different areas); NL3 (the internal audit has training policies for internal auditors); NL4 (continuous professional development activities are undertaken by internal auditors); NL5 (internal auditors annually arrange adequate short-term training courses for internal auditors); NL6 (the internal audit function has an adequate internal audit manual to guide the work of the internal audit); NL7 (the professional qualifications and training of internal auditors are at a higher level); NL8 (making organization decisions is based on valuable data sources and information from the internal audit); NL9 (training courses and the development of internal auditors are invested in consistently).

- HT (management support for internal audit) is measured by four observed variables: HT1 (the support from senior management for internal auditors is achieved to the auditors' expectations); HT2 (internal auditors' performed duties and responsibilities are supported by senior managers); HT3 (adequate organization budget enables the internal audit department to perform audit plan); HT4 (internal auditors receive sufficient support and encouragement from senior management for training and developing internal auditors).

- CL (the quality of the internal audit) is measured by four observed variables: CL1 (the annual audit plan is completely determined by the internal auditor); CL2 (the areas audited are very significant to the organization); CL3 (the internal audit can cover all organizational units and all issues); CL4 (regular follow-ups are performed by the internal auditors to examine recommendations taken to correct any problems found).

Accordingly, a questionnaire was designed which included both demographic questions as well as questions to investigate the impact of four independent variables on the dependent variable, using a five-level Likert scale (Score 1: Absolutely disagree; Score 2: Disagree; Score 3: Neutral; Score 4: Agree; Score 5: Absolutely agree).

In this study, the authors selected a sample of internal auditors working in companies listed on the Vietnamese stock market, using a convenience sampling method. The authors conducted a survey by sending a mail questionnaire directly to the internal auditors or directly calling the interviewees from May to November 2021. The size of the samples was determined to ensure the reliability of the results of exploratory factor analysis (EFA) and multivariate regression models. The minimum sample size used in exploratory factor analysis (EFA) should be more than 100 samples. The minimum sample size used in multivariate regression analysis is calculated by the formula is $50 + 8 \times 4 = 82$ (with four

independent variables in the model). Hence, 152 questionnaires were sent to satisfy both of these requirements.

Subsequently, the incomplete responses were removed from the study data; thus, the analyzed data consisted of 144 valid responses, and raw statistics were processed using SPSS software to test the reliability of the scale for variables measured by Cronbach alpha's index, perform exploratory factor analysis, derive descriptive statistics, and exploit regression analysis to evaluate the influence of independent variables on the effectiveness of internal audit of non-financial companies listed on the Vietnamese stock market.

## 4. Analysis of Results

### 4.1. Descriptive Statistical Analysis

General demographic information of the internal auditors participating in the survey is summarized in Table 1.

**Table 1.** Demographic information of internal auditors.

| Demographic Information of Internal Auditors | | Observation | Percentage (%) |
|---|---|---|---|
| Gender | Male | 54 | 37.5 |
| | Female | 90 | 62.5 |
| | Total | 144 | 100 |
| Working experience | <5 years | 22 | 15.2 |
| | >15 years | 4 | 2.9 |
| | from 11 years–15 years | 54 | 37.5 |
| | from 5 years–10 years | 64 | 44.4 |
| | Total | 144 | 100 |
| Working position | Internal auditor | 98 | 68 |
| | Deputy head of internal audit department | 28 | 19.4 |
| | Head of internal audit department | 18 | 12.5 |
| | Internal audit assistance | 4 | 0.1 |
| | Total | 144 | 100 |
| Education degree | Bachelor | 72 | 50.0 |
| | Master | 66 | 45.8 |
| | PhD | 6 | 4.2 |
| | Total | 144 | 100 |
| Professional Certificate | Yes (CPA, ACCA, CIA, CMA, FRM, … ) | 52 | 36.1 |
| | No | 92 | 63.9 |
| | Total | 144 | 100 |
| Internal audit department | Yes | 106 | 73.6 |
| | No | 38 | 26.4 |
| | Total | 144 | 100 |
| Internal audit function | Control Board | 12 | 8.3 |
| | Internal audit department (in-house) | 104 | 72.2 |
| | Combination between internal audit department and outsourcing a part of internal audit function | 22 | 15.3 |
| | Outsourcing 100% internal audit function | 6 | 4.2 |
| | Total | 144 | 100 |

Source: Results from the authors' study.

### 4.2. Reliability Analysis—Cronbach's Alpha

Table 2 shows the Cronbach's Alpha coefficient among the variables. The lowest Cronbach's alpha coefficient among the variables was 0.647 (higher than 0.6). Some observed variables, including L6, HL7, HL8, HL9, HL10, HL11, HL12, DL8, DL9, NL7, NL8, NL9, and HT4, had corrected item-total correlation lower than 0.4; thus, they were deleted from the model. Accordingly, only 25 observed variables were used to perform the subsequent EFA.

**Table 2.** Cronbach's alpha.

| Variables | Symbol | Number of Observed Variables | Cronbach's Alpha | Cronbach's Alpha If Item Deleted |
|---|---|---|---|---|
| **Dependent variables** | | | | |
| Internal audit effectiveness | HL | 12 | 0.647 | 0.870 (HL6, HL7, HL8, HL9, HL10, HL11, HL12) |
| **Independent variables** | | | | |
| Independence of internal audit | DL | 9 | 0.793 | 0.932 (DL8, DL9) |
| Competence of internal auditor | NL | 9 | 0.704 | 0.894 (NL7, NL8, NL9) |
| Management support for internal audit | HT | 4 | 0.788 | 0.845 (HT4) |
| Quality of internal audit | CL | 4 | 0.858 | |

Source: Results from the authors' study.

### 4.3. Exploratory Factor Analysis

Based on the sample, the calculated KMO coefficient was 0.754 (i.e., greater than 0.5). Thus, the sample size of the survey sample is eligible to conduct factor analysis. The result of Barlett's test shows that the $p$-value (Sig.) determined from the sample is 0.00 (i.e., less than 5%). Therefore, it is possible to reject hypothesis H0 (the correlation between observed variables is zero), or it can be concluded that the observed variables are correlated with each other overall.

To identify the main factors, the authors used a factor extraction method based on Eigenvalues. The analysis results calculated from the sample show that for the 25 items, 5 main factors could be extracted and explained 73.276% of the variation in the data set. The EFA results that identified the observed variables for each factor are shown in Table 3.

**Table 3.** Rotated Component Matrix[a].

| | Component | | | | |
|---|---|---|---|---|---|
| | **1** | **2** | **3** | **4** | **5** |
| DL5 | 0.902 | | | | |
| DL6 | 0.848 | | | | |
| DL3 | 0.837 | | | | |
| DL7 | 0.825 | | | | |
| DL2 | 0 | | | | |
| DL4 | 0.785 | | | | |
| DL1 | 0.732 | | | | |
| NL2 | | 0.924 | | | |
| NL1 | | 0.896 | | | |

**Table 3.** *Cont.*

| | Component | | | | |
|---|---|---|---|---|---|
| | **1** | **2** | **3** | **4** | **5** |
| NL5 | | 0.832 | | | |
| NL4 | | 0.754 | | | |
| NL6 | | 0.745 | | | |
| NL3 | | 0.618 | | | |
| HL3 | | | 0.814 | | |
| HL2 | | | 0.798 | | |
| HL5 | | | 0.779 | | |
| HL4 | | | 0.738 | | |
| HL1 | | | 0.631 | | |
| CL3 | | | | 0.876 | |
| CL2 | | | | 0.865 | |
| CL1 | | | | 0.827 | |
| CL4 | | | | 0.788 | |
| HT1 | | | | | 0.787 |
| HT3 | | | | | 0.774 |
| HT2 | | | | | 0.710 |

Source: Results from the authors' study.

### 4.4. Analysis of Descriptive Statistics

Here, the results of descriptive statistical analysis are presented in Table 4 as follows:

**Table 4.** Descriptive Statistics.

| Factors/Items | Minimum | Maximum | Mean | Std. Deviation |
|---|---|---|---|---|
| **Dependent variables** | | | | |
| **HL** **Effectiveness of internal audit** | **3** | **5** | **4.3444** | **0.45063** |
| HL1 | 3 | 5 | 4.4583 | 0.52908 |
| HL2 | 3 | 5 | 4.375 | 0.54223 |
| HL3 | 3 | 5 | 4.3472 | 0.58526 |
| HL4 | 3 | 5 | 4.25 | 0.52407 |
| HL5 | 2 | 5 | 4.2917 | 0.59191 |
| **Independent variables** | | | | |
| **F1** **Independence of internal audit** | **2.86** | **5** | **3.9921** | **0.58969** |
| DL1 | 1 | 5 | 4.0417 | 0.84649 |
| DL2 | 1 | 5 | 3.9444 | 0.7485 |
| DL3 | 2 | 5 | 3.9722 | 0.69144 |
| DL4 | 2 | 5 | 4.00 | 0.67135 |
| DL5 | 3 | 5 | 3.9722 | 0.64942 |
| DL6 | 3 | 5 | 3.9444 | 0.64762 |
| DL7 | 3 | 5 | 4.0694 | 0.61269 |

Source: Results from the authors' study.

### 4.5. Analysis of Regression Model

In order to evaluate the influence of independent factors identified in the exploratory factor analysis (EFA) on the effectiveness of internal audits, the authors conducted a regression analysis with the dependent variable as the effectiveness of internal audits of non-financial companies listed on the Vietnamese stock market. The results of correlation analysis between the independent and dependent variables are shown in Table 5:

**Table 5.** Correlations.

| | | HL | DL | NL | HT | CL |
|---|---|---|---|---|---|---|
| HL | Pearson Correlation | 1 | 0.439 ** | 0.179 | 0.517 ** | 0.323 ** |
| | Sig. (2-tailed) | | 0.000 | 0.132 | 0.000 | 0.006 |
| | N | 144 | 144 | 144 | 144 | 144 |
| DL | Pearson Correlation | 0.439 ** | 1 | 0.267 * | 0.432 ** | 0.301 * |
| | Sig. (2-tailed) | 0.000 | | 0.023 | 0.000 | 0.010 |
| | N | 144 | 144 | 144 | 144 | 144 |
| NL | Pearson Correlation | 0.179 | 0.267 * | 1 | 0.319 ** | 0.500 ** |
| | Sig. (2-tailed) | 0.132 | 0.023 | | 0.006 | 0.000 |
| | N | 144 | 144 | 144 | 144 | 144 |
| HT | Pearson Correlation | 0.517 ** | 0.432 ** | 0.319 ** | 1 | 0.357 ** |
| | Sig. (2-tailed) | 0.000 | 0.000 | 0.006 | | 0.002 |
| | N | 144 | 144 | 144 | 144 | 144 |
| CL | Pearson Correlation | 0.323 ** | 0.301 * | 0.500 ** | 0.357 ** | 1 |
| | Sig. (2-tailed) | 0.006 | 0.010 | 0.000 | 0.002 | |
| | N | 144 | 144 | 144 | 144 | 144 |

**, Correlation is significant at the 0.01 level (2-tailed). *, Correlation is significant at the 0.05 level (2-tailed). Source: Results from the authors' study.

The results of the correlation analysis showed that the independent variables statistically affected HL—the dependent variable—with a 1% significance level. However, the NL variable had a low correlation (0.179) with the significant value, of 0.132 > 0.05, indicating that the NL variable did not affect the dependent variable HL. Thus, in the next regression analysis, the NL variable was removed from the model. Regression analysis results are presented in Table 6 as follows:

**Table 6.** Statistical values of the regression model.

| Model Summary | | | | |
|---|---|---|---|---|
| Model | R | R Square | Adjusted R Square | Std. Error of the Estimate |
| 1 | 0.580 [a] | 0.336 | 0.307 | 0.37526 |

[a] Predictors: (Constant), CL, DL, HT. Source: Results from the authors' study.

The determination coefficient $R^2$ had a value of 0.336, which showed that the regression model with three main factors affected the effectiveness of the internal audit which can explain 33.6% of the variation in the effectiveness of internal audits of non-financial companies listed on Vietnam's stock market.

ANOVA results in Table 7 indicate the suitability of the regression model with an F value of 11.461 and *p*-value (Sig.) of 0.000 (which is less than the 5% significance level). Therefore, it can be concluded that the regression model is adequate to show the relationship between independent factors and the effectiveness of the internal audit.

**Table 7.** Analysis of variance—ANOVA.

| | Model | Sum of Squares | df | Mean Square | F | Sig. |
|---|---|---|---|---|---|---|
| | **ANOVA** [a] | | | | | |
| 1 | Regression | 4.842 | 3 | 1.614 | 11.461 | 0.000 |
| | Residual | 9.576 | 68 | 0.141 | | |
| | Total | 14.418 | 71 | | | |

[a] Dependent variable: HL. Predictors: (Constant), CL, DL, HT. Source: Results from the authors' study.

The evaluation of the impact of each factor on the effectiveness of the internal audit is shown by the regression coefficients, $\beta_i$, in the regression model. These factors were determined by the least-squares method; the calculation results from the survey sample are shown in Table 8.

**Table 8.** Coefficients [a].

| Model | | Unstandardized Coefficients | | Standardized Coefficients | t | Sig. | Collinearity Statistics | |
|---|---|---|---|---|---|---|---|---|
| | | B | Std. Error | Beta | | | Tolerance | VIF |
| 1 | (Constant) | 1.619 | 0.495 | | 3.271 | 0.002 | | |
| | DL | 0.187 | 0.085 | 0.244 | 2.196 | 0.032 | 0.789 | 1.267 |
| | HT | 0.368 | 0.113 | 0.369 | 3.250 | 0.002 | 0.757 | 1.321 |
| | CL | 0.116 | 0.106 | 0.117 | 1.094 | 0.278 | 0.846 | 1.182 |

[a] Dependent variable: HL. Source: Results from the authors' study.

However, to ensure the significance of the regression model, it is necessary to test assumptions about the meaning of the regression coefficient (if the regression coefficient is 0, the corresponding variable has no significance in the regression model). A pair of hypotheses was tested:

$$H0: \beta_i = 0 \tag{2}$$

$$H1: \beta_i \neq 0 \tag{3}$$

The test of this hypothesis is based on the probability value *p*-value (Sig.) to draw conclusions. Table 8 shows the *p*-values of each independent variable. The two independent factors (DL and HT) had *p*-values of 0.032 and 0.002, respectively, which are less than 0.005 significance level. Therefore, hypothesis H0 is rejected, and we can conclude that both DL (the independence of internal auditors) and HT (management support for internal audit functions) significantly impact on HL (the effectiveness of internal audits). Additionally, according to the results of this regression analysis, the coefficients $\beta_i$ of the two variables DL and HT both have positive signs, which proves that both these two independent variables have positive effects on the dependent variable (HL). The standardized regression coefficient of the independent variable HT is 0.369, which is higher than that of the independent variable DL, proving that the support from the managers will have a stronger impact on the effectiveness of the internal audit. In addition, the VIF coefficients of all independent variables were less than two, which proves that there is no multicollinearity among independent variables; therefore, the regression results of the model are reliable.

**5. Conclusions**

The results of the regression analysis show that the independence of an internal audit has a positive impact on the effectiveness of an internal audit, which is consistent with the results of Mutchler (2003); Alizadeh (2011); Baharud-din et al. (2014); and Al-Akra et al. (2016). Internal auditors have a right to unrestricted access to documents, personnel and other departments in collecting audit evidence, avoiding conflicts of interest between

internal auditors with other department, senior management, and the board of directors, which is similar to Goodwin and Yeo (2001) and Christopher et al. (2009). Therefore, hypothesis H1 is accepted.

The result also illustrated that management support for internal audit has a positive effect on the effectiveness of internal audits, which is in agreement with previous studies such as Albrecht et al. (1988, cited in Dellai and Omri 2016); Gramling et al. (2004); Salameh et al. (2011); Ahlawat and Lowe (2004); Aikins (2011); Octavia (2013); Baharud-din et al. (2014) and Dellai and Omri (2016). The analysis of the results pointed that the board of directors will support the approval of internal audit regulations and plans, appointment, resigning and salary decisions for internal auditors. For managers, support for internal audits in the administrative approval of personnel and allocated budget for internal auditors to implement the audit plan, and actively support the audit department in internal training and professional development. Hence, hypothesis H3 is accepted.

However, this research disagrees with Al-Twaijry et al. (2003); Alzeban and Gwilliam (2014); and Baharud-din et al. (2014) that the auditor's competence and quality of the internal audit have no effect on the effectiveness of the internal audit; thus, hypothesis H2 and hypothesis H4 are rejected.

In summary, this research systematized and determined the overview and characteristics of internal audits and presented a theoretical framework for factors affecting the internal audit effectiveness, including the independence of the internal auditor, competence of the internal auditor, management support for the internal audit, and the quality of the internal audit. This research verified the relationship between these factors and the effectiveness of the internal auditor in non-financial companies listed on Vietnam's stock market. However, the research has several limitations. Firstly, the research content focused on assessing the effectiveness of non-financial companies listed on the Vietnamese stock market. Secondly, the data used in this research's sample were relatively scant in comparison with previous studies examining internal audits. Therefore, the next research project may study factors affecting internal audit effectiveness of financial companies listed on the Vietnamese stock market and may use a larger sample size for analyzing the results.

**Author Contributions:** Conceptualization, T.T.T.; methodology, T.N.D.; software, T.N.D.; validation, T.T.T., T.N.D.; formal analysis, T.N.D.; investigation, T.T.T.; resources, T.T.T.; data curation, T.N.D.; writing—original draft preparation, T.T.T.; writing—review and editing, T.N.D.; visualization, T.N.D.; supervision, T.T.T.; project administration, T.N.D. All authors have read and agreed to the published version of the manuscript.

**Funding:** This research received no external funding.

**Institutional Review Board Statement:** Not applicable.

**Informed Consent Statement:** Not applicable.

**Data Availability Statement:** The datasets used and/or analyzed during the current study are available from the corresponding author on reasonable request.

**Conflicts of Interest:** The authors declare no conflict of interest.

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
