# Peer review of "Factors Affecting Internal Audit Effectiveness: Empirical Evidence from Vietnam"

_ijfs, doi:10.3390/ijfs10020037_

Round 1
Reviewer 1 Report
The article is very interesting as well as current. However, it has a number of shortcomings that I believe need to be addressed. The authors primarily deal with the issue of audit effectiveness in the stock market. It would also be appropriate to use the knowledge of some European authors who also deal with this issue and whose work brings new valuable knowledge. An example is
Mucha, B. (2021). Evaluation of the State of Implementation of the European Structural and Investment Funds: Case Study of the Slovak Republic. Online Journal Modeling the New Europe, 35, 4-24, doi: 10.24193 / OJMNE.2021.35.0
The authors also focus on the place of research, which is the Vietnamese stock market. However, there is at least a brief definition of the stock market, as it does, for example
Peracek, T. (2020). Legal issues and practical problems in the management of the securities portfolio in Slovakia. AD ALTA-Journal of Interdisciplinary research, 10 (2), pp. 255-260
Due to the smaller number of references, it is appropriate to expand the article with other resources dealing with the researched issues, such as:
Strakova, J .; Korauš, A .; Váchal, J .; Pollak, F .; Cernak, F .; TalíÅ™, M .; Kollmann, J. Sustainable Development Economics of Enterprises in the Services Sector Based on Effective Management of Value Streams. Sustainability 2021, 13, 8978. https://doi.org/10.3390/su13168978
Pavelek, O., Zajickova, D. (2021). Personal Data Protection in the Decision-Making of the CJEU before and after the Lisbon Treaty.
TalTech Journal of European Studies, 11 (2), pp. 167–188, doi: 10.2478 / bjes-2021-0020
The authors set out four hytotheses in the article. The article lacks answers to some of them. In conclusion, it would be appropriate to unequivocally confirm or reject all established hypotheses.
At the end of the article, it is appropriate to state at least a few sentences concerning the authors' future research
Author Response
Point 1: The authors primarily deal with the issue of audit effectiveness in the stock market. It would also be appropriate to use the knowledge of some European authors who also deal with this issue and whose work brings new valuable knowledge. An example is Mucha, B. (2021). Evaluation of the State of Implementation of the European Structural and Investment Funds: Case Study of the Slovak Republic. Online Journal Modeling the New Europe, 35, 4-24, doi: 10.24193 / OJMNE.2021.35.0
Response 1:
The authors have add the research of Mucha, B. (2021) into “Introduction” part and references.
Point 2: The authors also focus on the place of research, which is the Vietnamese stock market. However, there is at least a brief definition of the stock market, as it does, for example
Peracek, T. (2020). Legal issues and practical problems in the management of the securities portfolio in Slovakia. AD ALTA-Journal of Interdisciplinary research, 10 (2), pp. 255-260
Response 2:
The authors have add a brief definition of stock market and introduction of Vietnamese stock market in to “Introduction” part.
Point 3: Due to the smaller number of references, it is appropriate to expand the article with other resources dealing with the researched issues, such as:
Strakova, J .; Korauš, A .; Váchal, J .; Pollak, F .; Cernak, F .; TalíÅ™, M .; Kollmann, J. Sustainable Development Economics of Enterprises in the Services Sector Based on Effective Management of Value Streams. Sustainability 2021, 13, 8978. https://doi.org/10.3390/su13168978
Pavelek, O., Zajickova, D. (2021). Personal Data Protection in the Decision-Making of the CJEU before and after the Lisbon Treaty. TalTech Journal of European Studies, 11 (2), pp. 167–188, doi: 10.2478 / bjes-2021-0020
Response 3:
The research of Strakova et al. (2021) is added into Part 2.3, and reference.
The research of Pavelek and Zajickova (2021) is added into “Introduction” part and reference
Point 4: The authors set out four hytotheses in the article. The article lacks answers to some of them. In conclusion, it would be appropriate to unequivocally confirm or reject all established hypotheses.
Response 4:
The authors confirmed the hypothesis H1 and hypothesis H3, rejected hypothesis H2 and hypothesis H4 which are presented in Conclusion.
Point 5: At the end of the article, it is appropriate to state at least a few sentences concerning the authors' future research
Response 5:
The author added the limitation of research and future research at the end of the article.
Reviewer 2 Report
Dear Authors,
thank you very much for your efforts and interest in internal audit. The topic you focused on, is really very important for many entrepreneurs and economic entities, especially if we consider efficiency of this process. However, I think your work needs a little improvement. My suggestions and comments are included in the attached file.
Best regards

Author Response
Point 1: Introduction: How do you understand effectiveness of internal audit? If this is a subject of research, you should explain how it is explained, what the authors understand under this construct? How it is measured?
Response 1:
- The definition of effectiveness of internal audit is added to “Introduction” part.
- The method to measure effectiveness of internal audit is provided in Part 3. Research Methodology
Point 2: Hypothesis 1 & Hypothesis 2: If there exist the research of the connection between independence and the effectiveness of internal audit and they suggest/proof this positive relation, what is the novelin your research? what is different/new in this hypothesis according to previous research?
Response 2:
- The Hypothesis 1 & 2 have been based on the previous researches such as Mutchler (2003), Baharud-din et al. (2014), Al-Akra et al. (2016) for indenpendece of internal audit afftecting positively effectiveness of internal audit and Al-Twaijry et al. 2003; Alzeban & Gwilliam 2014, Mihret & Yismaw 2007; Onumah & Yao Krah 2012; Walter & Guandaru 2012 for the competence of internal auditors affecting positively effectiveness of internal audit. The main difference between the previous studies and our researh is that Vietnam is a developing country and the new Decree 05/2019/ND-CP of Vietnamese Governance related to internal audit took effect on April 1, 2019 and actually started to apply compulsory internal audit function in listed companies on April 1, 2021. Thus, compared with previous studies which are developed contries or developing contries but they have long time to implement mandatory requirement of the internal audit function in listed companies. The authors have included information about this new regulation of Vietnam in the Introduction section
- In addition, in “Conclusion” part, the paper has been pointed out the the results of research that the independence of internal audit and the management support for internal audit affecting positively effectiveness of internal audit consisted with the results of Mutchler (2003), Alizadeh (2011), Baharud-din et al. (2014), Al-Akra et al. (2016), ….. However, this research disagree with Al-Twaijry et al. (2003), Alzeban & Gwilliam (2014) and Baharud-din et al. (2014) that the auditor’s competence and quality of internal audit work have no effect on the effectiveness of internal audit
Point 3: Research Methodology: there were observations or the declarations? What exactly you ask respondents to assess this variables? You didn't explain clearly the structure of the questionnair, when the survey was make and what method was used.
Response 3:
- Dependent variable and independent variables are measured by observed variables. Therefore, the authors changed “observation” to “observed variable”
- The authors have added the content of the observed variables, corresponding to the content the authors ask in questionnaire to assess these dependent and independent variables.
- The structure of the questionnaire is added to “Research Methodology” part.
- The time and the method of the survey is added.
Point 4: Research Methodology: I think you should start with explaining the variables and how they were measured and why. Than you can describe the model and used tools for testing
Response 4: The authors have revised the "Research methodology" section according to the reviewer's suggestions.
Point 5: Research Methodology: How did you aggregate this observations to one variable?
Response 5:
- The authors have inherited previous research (Goodwin & Yeo, 2001; Mutchler, 2003; Christopher et al., 2009; Aaron & Gabriel, 2010; Walter & Guandaru, 2012; Baharud-din et al., 2014; Alzeban & Gwilliam, 2014; Dellai & Omri, 2016; Al-Akra et al., 2016) and combined with depth interview to select items for variable and group into a variable.
- The authors added this explanation in the "Research methodology" section.
Round 2
Reviewer 1 Report
I recommend accepting the article for publication in the current modified version.
Reviewer 2 Report
Dear Author(s),
thank you very much for your response. I think your corrections are satisfy.